# Mobile Robot Localization and Mapping Algorithm Based on the Fusion of Image and Laser Point Cloud

**DOI:** 10.3390/s22114114

**Published:** 2022-05-28

**Authors:** Jun Dai, Dongfang Li, Yanqin Li, Junwei Zhao, Wenbo Li, Gang Liu

**Affiliations:** School of Mechanical and Power Engineering, Henan Polytechnic University, Jiaozuo 454003, China; 211905010028@home.hpu.edu.cn (D.L.); zjwmail@hpu.edu.cn (J.Z.); 212005010031@home.hpu.edu.cn (W.L.); 212005010032@home.hpu.edu.cn (G.L.)

**Keywords:** SLAM, nonlinear optimization, multi-sensor fusion, state estimation

## Abstract

Given the lack of scale information of the image features detected by the visual SLAM (simultaneous localization and mapping) algorithm, the accumulation of many features lacking depth information will cause scale blur, which will lead to degradation and tracking failure. In this paper, we introduce the lidar point cloud to provide additional depth information for the image features in estimating ego-motion to assist visual SLAM. To enhance the stability of the pose estimation, the front-end of visual SLAM based on nonlinear optimization is improved. The pole error is introduced in the pose estimation between frames, and the residuals are calculated according to whether the feature points have depth information. The residuals of features reconstruct the objective function and iteratively solve the robot’s pose. A keyframe-based method is used to optimize the pose locally in reducing the complexity of the optimization problem. The experimental results show that the improved algorithm achieves better results in the KITTI dataset and outdoor scenes. Compared with the pure visual SLAM algorithm, the trajectory error of the mobile robot is reduced by 52.7%. The LV-SLAM algorithm proposed in this paper has good adaptability and robust stability in different environments.

## 1. Introduction

Mobile robots are widely used globally. Especially after the outbreak of the epidemic, the use of “non-contact” robots has been increasing rapidly. SLAM (simultaneous localization and mapping) is the primary technology to complete the positioning and mapping of the robot, which is the premise of realizing the autonomous navigation of the robot [1,2,3]. The visual SLAM algorithm has poor robustness, low positioning accuracy, and drift phenomenon when the mobile robot has been moving for a long time, which cannot guarantee its long-term stable operation in various environments. To improve the robustness and accuracy of the visual SLAM algorithm, the laser point cloud data are integrated to complete the work of positioning and mapping.

The visual SLAM algorithm mainly obtains environmental information through a camera sensor and calculates its position and surrounding environment according to the collected images. The visual SLAM algorithm can be divided into the feature point method [4], the direct method [5,6,7], and the semi-direct method [8] according to whether to perform feature extraction and description. According to the different visual sensors, it is also mainly divided into monocular, stereo, and RGB-D solutions. If a monocular camera is used, the scale of motion is often unsolvable without the help of other sensors or assumptions about motion [9,10,11]. A stereo camera determines the motion scale through the baseline, but the algorithm is complex and requires high computational performance [12]. The introduction of RGB-D cameras provides an efficient method for visually correlating images with depth [6,13]. Motion estimation can be performed on a large scale using RGB-D cameras. Still, these methods only use the imaging regions with depth, which may waste most of the areas in the visual image that are not covered by depth. RGB-D cameras are easily affected by lighting, which is unusable in outdoor environments. Mur-Artal et al. [14,15] suggested the ORB-SLAM algorithm based on PTAM, using three threads of tracking, mapping, and loopback to achieve SLAM, and introduced the concepts of con-view and essential mapping to describe the constraint relationship between frames. The bag of words (BoW) model is used for loop closure detection and relocation [16]. The effect of eliminating the cumulative error of system positioning is noticeable. The ORB-SLAM2 algorithm can achieve sparse mapping and high-precision localization, but it is very dependent on the quantity and quality of features detected in the environment, and the tracking is unstable. Greter et al. [17] used lidar-assisted cameras to implement SLAM, which focused on the selection of features and keyframes, and adopted a variety of strategies to improve the quality of input data, such as using laser point clouds to segment image features on the ground, processing a feature separately according to the distance between the feature and the vehicle body, eliminating the outliers by semantic markup, and approaching the plant features separately. Still, the algorithm complexity is high, which requires a GPU to achieve real-time performance and mark semantic information in advance.

Zhang et al. [18] combined laser and RGB-D sensors with a filter-based method to realize the SLAM algorithm. The probabilistic heuristic model is used to extract the beam projection to the grid map unit, making full use of the redundant information in the laser and visual to perform feature-level information fusion. In the map update stage, the Bayesian estimation method in fusing the data of laser and visual sensors to update the grid map is utilized, which effectively improves the accuracy and robustness of the SLAM algorithm. For the SLAM algorithm of the visual sensor and lidar fusion, to make full use of the depth information provided by lidar, Qi et al. [19] improved the LIMO algorithm, which implemented accurate localization by introducing a multi-strategy fusion mechanism. The depth value of the feature in the previous frame is judged, estimating the pose based on depth information classification of the feature, which effectively improves the positioning accuracy and robustness.

Vision-based methods can easily extract environmental features, but visual SLAM algorithms are susceptible to changes in lighting and viewpoints. If the camera is used as the only navigation sensor, the adaptability to the environment is poor, and the system cannot run stably. In addition, the image collected by the monocular camera lacks scale information. The accumulation of many features lacking depth information will cause scale blur, which will lead to degradation and tracking failure, especially when the camera moves along the optical axis, although the feature can still be detected by the visual sensor, and then the depth information of the feature point can be obtained by triangulation to restore the rotational motion. Since the visual feature may only move very little, it is difficult to estimate the translational motion of the camera, and the final motion estimation is degraded. The lidar-based SLAM algorithm works well even at night, with a wide scanning range and high accuracy, and the high-resolution 3D lidar can capture environmental details at longer distances. Therefore, this study proposes a localization and mapping method, LV-SLAM, which uses the data of the laser point cloud to provide additional depth information for image features to assist the monocular camera in completing motion estimation and uses a keyframe-based method to optimize the pose locally. At the same time, the point cloud is processed and superimposed according to the estimated high-precision pose transformation to achieve dense mapping. Compared with the LIMO algorithm, our method not only improves the quality of the input information of the system, but it also pays attention to the correlation between frames and at the same time adds more constraint information to the system. We experimented with the proposed method on real scenarios and KITTI datasets [20], which is currently the largest computer vision algorithm evaluation dataset in the world for autonomous driving scenarios. The results of the KITTI dataset experiments and outdoor real-world experiments show that the improved algorithm has stable tracking and strong adaptability to the environment and can obtain higher-precision pose estimation and dense mapping effects.

## 2. Materials and Methods

The visual SLAM algorithm can output a relatively accurate motion trajectory and map in a short period. Still, due to the accumulation of errors, the motion trajectory gradually diverges over a long time, and the map is no longer credible. For the mobile robot to maintain the optimal trajectory for a long time, the robustness of the algorithm must be enhanced, so that the mobile robot can still operate efficiently and stably in harsh environments or during vigorous exercise. The visual SLAM algorithm uses feature points to complete positioning, and the set of feature points is the constructed sparse point cloud map. However, as the underlying technology, SLAM must build a dense map of the environment to meet related tasks such as navigation, obstacle avoidance, and reconstruction. Based on visual SLAM, this paper proposes an approach to fusing image and laser point clouds for SLAM. The algorithm framework is shown in Figure 1.

Firstly, the external parameters of the camera and lidar sensor are calibrated. The laser point cloud is projected into the image coordinate system using the calibrated external parameter information. The ORB features are extracted from each frame. In the laser vision fusion module, the image features are combined with the laser points to obtain the feature depth information to complete the initialization. Then, the perspective-n-point (PNP) algorithm is used to solve the initial value of the inter-frame motion and utilize the Levenberg–Marquardt (LM) method for iterative optimization, which help to triangulate the remaining feature without depth information. After the above steps, bundle adjustment (BA) optimization will be performed on all feature points and poses to output the optimized poses at a low frequency. The keyframe is identified by the previously estimated motion and the number of feature points. The place recognition is the process of similarity detection between keyframes of the input image given the dictionary made in advance. The estimated motion is corrected after determining the loop closure. In the motion estimation module, the odometer constraints, loopback constraints, and global pose optimization are unified to obtain a high-precision camera pose. The dense map can be obtained by processing and superimposing the point cloud according to the high-precision pose transformation estimated in the previous step. The camera pose is estimated and optimized through augmented image features, and the map is derived from a laser point cloud.

### 2.1. BA-Based Back-End Optimized Visual SLAM Algorithm

Due to the presence of noise, all detected feature points and solved poses must not fully satisfy the observation equation. The essence of the optimization algorithm is to estimate the state parameters as accurately as possible in the noisy data, making it approximately fit the equation. The back-end includes an optimization thread for local keyframes, which is responsible for smoothing the camera pose and creating a more consistent map. In the iterative solution, the features and poses in the state space are selected as the variables to be optimized. Then, the optimal solution of the objective function is found in the corresponding gradient descent direction. Even if the initial value of motion estimation deviates significantly, the camera trajectory can be quickly and accurately recovered by an iterative solution.

#### 2.1.1. Visual Odometry

Visual odometry (VO) is the front-end of the visual SLAM algorithm, which mainly calculates the robot’s pose through images. The camera collects environmental images at a certain frequency during motion, and the image collected at time *k* is denoted as $I_{k}$. The image sequence acquired in period *K* is denoted as $I_{0:n}=\left\{I_{0},I_{1},\cdots ,I_{n}\right\}$. According to the pose of the camera in the initial state $T_{0}$, the pose of the camera at the moment *k* can be expressed as Equation (1):(1)$$T_{k}=T_{k,k-1}T_{k-1}=\left[\begin{array}{cc} R_{k,k-1} & t_{k,k-1}\\ 0 & 1 \end{array}\right]T_{k-1}$$
where $T_{k,k-1}$ is the transformation matrix of the rigid body from frame $I_{k-1}$ to frame $I_{k}$, $R_{k,k-1}\in so\left(3\right)$ is the rotation matrix, and $t_{k,k-1}\in R^{3\times 1}$ is the translation vector. The set $T_{0:k}=\left\{T_{1,0},T_{2,1},\cdots ,T_{k,k-1}\right\}$ contains all subsequent movements. VO solves the relative motion transformation between the frame $I_{k-1}$ and $I_{k}$. Transformations are then connected to restore the full camera trajectory.

The camera pose can be recovered incrementally by Equation (1). Due to inevitable accumulated errors and noise, the camera trajectory will drift with the long-term and long-distance operation of the mobile robot. Bundle adjustment optimization (BA) obtains more accurate and robust motion estimation [21].

#### 2.1.2. BA-Based Back-End Optimization

The method based on graph optimization considers all the observation information to give a more consistent camera pose. As shown in Figure 2, in the local bundle adjustment module, the image features are projected into other image frames, and the reprojection error is used to construct an objective function to optimize the pose and 3D feature points of the mobile robot, converting the state estimation problem into a nonlinear least-squares problem. The global optimization is to retain all the image frame information and regard the inter-frame transformation as a random variable and optimize the robot pose through the maximum likelihood estimation criterion.

The motion estimation between three-dimensional (3D) world points and two-dimensional (2D) image feature points is called the PNP problem [22]. Three-dimensional space points are selected from the reference image frame and projected to the current image frame to obtain 2D projection points. The PNP problem can also be solved using a nonlinear optimization approach by minimizing the reprojection error. The reprojection error model is shown in Figure 3. The observation points of the spatial feature point *p* in the two frames of images before and after are ${\tilde{p}}_{k}$ and ${\tilde{p}}_{k-1}$, and the projection point ${{\tilde{p}}^{\prime }}_{k-1}$ is obtained when the currently estimated pose is projected to the frame k−1 image. The error e is obtained by comparing ${{\tilde{p}}^{\prime }}_{k-1}$ with ${\tilde{p}}_{k-1}$.

The least-squares problem is constructed according to the reprojection error, and the optimal pose is solved by minimizing the reprojection error:(2)$$\xi^{*}=\underset{\xi }{\arg\min}\frac{1}{2}\sum_{i}^{n}||{\tilde{P}}_{i}-\pi (K,\xi )P_{i}|{|}_{2}^{2}$$
where ξ represents the motion transformation between two frames of images, i is the number of feature points in the image, $P_{i}$ represents the 3D point in space, ${\tilde{P}}_{i}$ represents the coordinates of the image observation point, and π(⋯) is the projection function from the 3D world point to the 2D image point.

BA reduces the global objective function by continuously adjusting the state quantity of the system until the algorithm converges to obtain the optimal solution. The rotation matrix described in Euclidean space contains orthogonal constraints, and the determinant is 1$\left(RR^{T}=I,D\mathrm{et}\left(R\right)=1\right)$, which seriously affects the convergence efficiency of optimization. If the transformation matrix of 16 variables is used to express the motion of the mobile robot with 6 degrees of freedom, the redundancy increases. To reduce the difficulty of optimization, Lie algebra is used to represent the camera pose, and the optimization problem is transformed into a general unconstrained nonlinear least-squares question. As a nonlinear system, SLAM is difficult to directly derive the pose of the objective function. Therefore, using the perturbation model, the optimal pose is indirectly calculated by solving the pose increment. In the objective function of the global BA, the variables to be optimized are put together for optimization in batches: $x={\left[\xi_{1},\cdots ,\xi_{m},p_{1},\cdots ,p_{n}\right]}^{T}$, Δx is the increment of the system independent variable. $\Delta x^{*}$ can be solved by minimizing the objective function.
(3)$$\begin{array}{cc} \Delta x^{*} & =\underset{\Delta x}{\arg\min}\frac{1}{2}||\left(f(x+\Delta x)\right)|{|}^{2}\\ & \approx \underset{\Delta \xi ,\Delta p}{\arg\min}\frac{1}{2}\sum_{i=1}^{m}\sum_{j=1}^{n}||e_{ij}+F_{ij}\Delta \xi_{i}+E_{ij}\Delta p_{j}|{|}^{2}\\ & =\underset{\Delta x}{\arg\min}\frac{1}{2}\left(||f(x)|{|}_{2}^{2}+2f{\left(x\right)}^{T}J(x)\Delta x+\Delta x^{T}H(x)\Delta x\right) \end{array}$$
where $e_{ij}$ represents the residual in the state, i represents the i-th frame image, j represents the number of feature points on a single frame image, ξ is the Lie algebra form of the robot pose, p represents the feature points in the map, and *F* and *E* represent the entire objective function in the partial derivative of the camera pose and the position change of the feature point in the current state. $x+\Delta x^{*}$ near the local optimum $x^{*}$ is the optimal solution, and the derivative of the loss function about Δx is 0. Formula (3) can be derived to obtain Equation (4).
(4)$$H(x)\Delta x=-J{\left(x\right)}^{T}f(x)$$
where *J* and *H* are the first-order and second-order derivative matrices of the residual function to the variable, respectively. In the solution process, the loss function gradually decreases as the number of iterations increases. The derivation problem becomes a process of constantly finding the gradient descent direction (JΔx<0) until the increment of the independent variable is very small and the objective function converges to a minimum value. The Gauss–Newton (GN) algorithm [23] is used to solve the increment Δx, where $H=J^{T}J$. The specific solution steps are as follows:
In the motion estimation, the initial value of the pose is obtained by matching frames.For K-th iterations, find the Jacobian matrix $J(x_{k})$ of the current state and the residual of the objective function $f(x_{k})$. Find the increment $\Delta x_{k}$, so that the objective function reaches a minimum value until $F(x_{k+1})<F(x_{k})$.Set the threshold, if $\Delta x_{k}$ is less than the threshold, to stop the iteration and set $x_{k}=x_{k}+\Delta x_{k}$.Otherwise, set $x_{k+1}=x_{k}+\Delta x_{k}$, return 2.


### 2.2. Pixel-Level Fusion of Images and Point Clouds

This paper proposes a method of using laser point cloud data to provide additional depth information for image features to assist the monocular camera to complete motion estimation. Even if the point cloud is relatively sparse, it can also complete the auxiliary work. Figure 4 shows the process of correlating image features with depth information of laser cloud.

In Figure 4, the distance between the normalized plane and the camera optical center is denoted as L. The three laser points closest to the image feature points in the normalized plane are denoted as $p_{L^{\prime }}^{1}$, $p_{L^{\prime }}^{2}$, $p_{L^{\prime }}^{3}$, and the corresponding spatial laser points are denoted as $p_{L}^{1}$, $p_{L}^{2}$, $p_{L}^{3}$. The gray plane is a local plane constructed by the three spatial laser points. The yellow point $p_{L}$ is the intersection of the line passing through the camera optical center and the image feature points with the laser point fitting plane.

First, the laser point cloud is unified into the camera coordinate system, and then the laser point cloud is preprocessed, and only the laser points in the front view of the camera are retained. The laser point cloud at the same time is projected onto the image with the latest timestamp, and the laser points are down-sampled to ensure that the point cloud can be evenly distributed. For each image feature point $p_{C}$, the feature point and the laser point are projected onto a normalized plane at a distance of 10 m from the optical center of the camera, and then for each image feature point we find the three nearest lidar projection points $p_{L^{\prime }}^{J}\;$, J∈{1,2,3} on the normalized plane. The three spatial laser points $p_{L}^{J}\;$, J∈{1,2,3} corresponding to the lidar projection points can form a local plane in the 3D space, and the 3D feature points $p_{L}$ can be obtained by projecting the image feature points along the optical axis to this plane. The depth of the feature points can be calculated by the Equation (5).
(5)$$\left\{ \begin{array}{c} (p_{L}-p_{L}^{1})\left(\left(p_{L}^{1}-p_{L}^{2})\times (p_{L}^{2}-p_{L}^{3}\right)\right)=0\\ (p_{L}-p_{L}^{2})\left(\left(p_{L}^{1}-p_{L}^{2})\times (p_{L}^{2}-p_{L}^{3}\right)\right)=0\\ p_{L}\in L_{S_{o}p_{c}} \end{array}\right.$$

In practical applications, due to the short time between image frames and the limited movement distance of the robot, point clouds may accumulate. Due to environmental occlusion and other reasons, image feature points may be associated with point clouds on other objects, resulting in depth blurring, such as what is shown in Figure 5.

In Figure 5, the blue point in the figure represents the laser point detected by the camera at moment $t_{i}$, and the yellow point represents the laser point that the camera moves to a new position and observes at moment $t_{j}$. However, due to the accumulation of point clouds, the blue points in dotted line observed at moment $t_{i}$ are also treated as being observed at time $t_{j}$. Using point clouds of other objects to estimate the feature depth may reduce the accuracy of motion estimation. After determining the three laser spots around the feature, the feature depth will be verified by examining the depth distance between the laser spots. If the maximum distance is greater than *d* customized, it is considered that this feature is not associated with the laser point. The feature will be processed without depth information.

The above method does not associate all image features with a single laser point but first filters the image features and only uses this method for depth estimation when the laser points around the image features meet the conditions, which can give full play to the complementary advantages of the sensor. This paper aims to exploit sparse depth information from lidar to solve motion involving features with or without depth information. The depth information is still useful for visual features even if the point cloud is sparse.

### 2.3. Pose Estimation Based on the Improved Objective Function

Visual SLAM algorithms based on image features are suitable for urban scenes with rich structural information. However, in situations where the number of structures is low and the optical flow is large, such as playgrounds and open roads, the number of extracted features is limited, and the value of depth may be too small to obtain accurate motion estimation. To solve the problem that few features have effective depth, the epipolar error was introduced, which strengthens the stability of the algorithm by augmenting constraint in features. Assume that the set of feature points detected at runtime is *I*, and the correspondence of feature points $X_{i}^{k-1}$ and $X_{i}^{k}$ detected from frame k−1 and frame k can be expressed as Equation (6).
(6)$${\bar{X}}_{i}^{k-1}K^{-T}t^{\wedge }RK^{-1}{\bar{X}}_{i}^{k}=0$$
where *R* and *t* represent the rotation matrix and translation vector of the mobile robot motion, respectively. When the *k* frames of images are processed, the laser point cloud of the *k*-1 frame has been obtained, then $X_{i}^{k-1}$ has depth information, and the pixel coordinate $X_{i}^{k}$ is denoted as ${\bar{X}}_{i}^{k}$. According to Equation (2), the reprojection relationship can be expressed as Equation (7).
(7)$${\bar{X}}_{i}^{k}=K(RX_{i}^{k}+t)$$

Constraints can be provided to the system whether or not features have depth information by correlating laser point clouds or triangulation. We take the two-norm of the residuals brought by all matching errors as the overall objective function:(8)$$\underset{R,t}{\arg\min}\sum_{i}\left(\rho_{3D-2D}\left(||{\bar{X}}_{i}^{k}-K(RX_{i}^{k-1}+t)|{|}_{2}^{2}\right)+\rho_{2D-2D}\left(||{\bar{X}}_{i}^{k-1}K^{-T}t^{\wedge }RK^{-1}{\bar{X}}_{i}^{k}|{|}_{2}^{2}\right)\right)$$

The improved objective function is used for inter-frame motion estimation, and the algorithm flow is shown in Algorithm 1.
**Algorithm 1:** Frame-to-Frame Motion Estimation
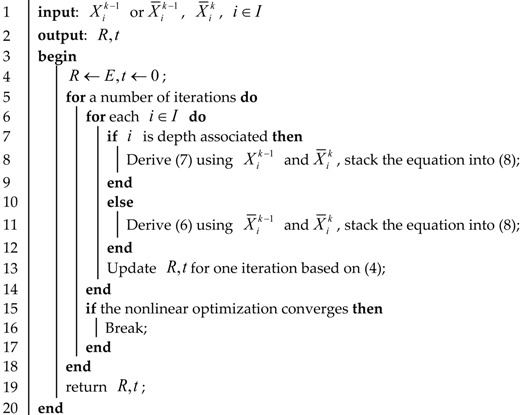


In the motion estimation algorithm, the features in the frame k-1 are divided into $X_{i}^{k-1}$ and ${\bar{X}}_{i}^{k-1}$ for the features with and without depth. The pose of mobile robot is initialized on line 4. The matching residuals are calculated by the Equations (6) and (7) and then stacked into the overall objective function. On line 13, the GN optimization algorithm is used to iteratively solve Equation (8) until the algorithm convergence is found. Finally, the motion transformation R,t from frame k−1 to frame k is returned.

### 2.4. Keyframe-Based Local Optimization

Taking all feature points and camera poses as parameters to be estimated, the optimization problem becomes very large and takes up huge computing resources. To achieve online estimation, it is necessary to reduce the complexity of the optimization problem. The sliding window can be used to reduce the number of parameters by removing previous feature points and poses. Though the local error is reduced, the global drift is not improved. Therefore, the paper uses a keyframe-based method to locally optimize the pose.

To improve the processing efficiency, only a certain number of keyframes are selected for optimization. When a new keyframe l is selected, according to the feature points of the current frame, find the top nine local keyframes with the highest degree of correlation. The ten keyframes are formed into an image sequence J and sorted according to the degree of association with the frame l. In addition, l is set as the first frame in the sequence; j are the other image frames in the sequence except the first frame. After initialization, all feature points in the local keyframe are projected to the current keyframe, and the objective function of local BA is constructed according to the reprojection error and epipolar error. This paper implements BA optimization with the help of the open library Cere. The different types of features can be easily processed by customizing the variables to be estimated and residual items. Redefine the feature points in the frame *K* as ${\tilde{X}}_{i}^{K}={\left[{\bar{x}}_{i}^{k},{\bar{y}}_{i}^{k},z_{i}^{k}\right]}^{T}$, ${\bar{x}}_{i}^{k}$, ${\bar{y}}_{i}^{k}$, which represents the normalized coordinates of the feature, $z_{i}^{k}$ represents the depth of the feature, and $z_{i}^{k}$ is set to the default value for features without depth. The local BA optimization can be expressed as:(9)$$\underset{T}{\arg\min}\sum_{i,j}{\left(T_{l}^{j}{\tilde{X}}_{i}^{l}-{\tilde{X}}_{i}^{j}\right)}^{T}\Omega_{i}^{j}(T_{l}^{j}{\tilde{X}}_{i}^{l}-{\tilde{X}}_{i}^{j})i\in I,j\in J\backslash \{l\}$$
where ${\tilde{X}}_{i}^{j}$ represents the observed value of the feature i in the *j*-th frame, $\Omega_{i}^{j}$ represents the information matrix, and $T_{l}^{j}$ is the transformation matrix from the *j*-th frame to the *l*-th frame in the sequence. The third row of the information matrix parameter depends on the depth source of the image feature; if the depth of the feature originates from the laser A point cloud, it will be assigned a larger value. the depth of this image feature is calculated by triangulation, given a smaller value, and is inversely proportional to the depth value, if no depth is set to be 0.

After being optimized by local BA, the refined pose is combined with the pose estimated by the previous odometry. The frequency of outputting the pose of the mobile robot is consistent with the speed of the pose estimation between frames. Keyframe-based local optimization speeds up algorithm processing while further improving accuracy and stability.

## 3. Results

The experimental platform used in this paper is shown in Figure 6. The experimental platform is a four-wheel skid steering mobile robot developed by Hefei Zhongke Shengu Technology Development Company. It is jointly developed based on MATLAB and ROS and equipped with a Linux operating system and an ROS Version 18.04.6 (Melodic) control system.

The mobile robot is equipped with LS-C16 lidar, an Intel D435 camera, the system controller model is an NVIDIA Jetson AGX Xavier, and the chassis is equipped with a 24 V lithium battery, which provides stable power for sensors and actuators.

### 3.1. KITTI Dataset Experiment

#### 3.1.1. Motion Trajectory and Mapping Experiment

The 00-10 sequences of the KITTI dataset provide the real movement trajectory during the movement process. The algorithm performance can be verified by comparing the movement trajectory solved by the fusion SLAM algorithm with the real trajectory. To verify the positioning accuracy and adaptability of the fusion SLAM algorithm in different types of environments, the 00 sequence, 03 sequence, and 09 sequence of the KITTI dataset are selected as the experimental scenes. The output results of the fusion algorithm LV-SLAM are compared with the laser SLAM algorithm A-LOAM and the visual SLAM algorithm ORB-SLAM2. The results of the three algorithms in the 00 sequence, 03 sequence, and 09 sequence are shown in Figure 7.

The left side of Figure 7 shows the trajectory comparison results of the three algorithms in different scenarios. Here, the dotted line is the real trajectory; the blue, green, and red lines are the output results of the ORB-SLAM2, A-LOAM, and LV-SLAM algorithms, respectively. Among them, the 00 sequence contains a large number of buildings, which are enough stable features. The tracking is relatively stable. The output motion trajectory of the three algorithms is consistent with the real trajectory as a whole. However, compared with the visual SLAM algorithm, the output results of the fusion algorithm and the laser SLAM algorithm are closer to the real trajectory. There are many trees on both sides of the road in the 03 sequence. A large number of image features can be detected in both the close and distant scenes. The output motion trajectory of the fusion algorithm, the visual SLAM algorithm, and the laser SLAM algorithm almost coincide with the real trajectory, and the differences between the various trajectories are small. In the 09 sequence with obvious scene changes, vegetation and buildings appear alternately, and structural features and natural features are mixed. The motion estimation deviation of the vision-based ORB-SLAM2 algorithm is relatively large, and the trajectory drift phenomenon becomes obvious as the driving distance increases. Especially in some scenarios (vehicle turning), the number of features detected by the ORB-SLAM2 algorithm plummeted, and even the tracking was lost, while the LV-SLAM algorithm and the A-LOAM algorithm were not affected.

Overall, in the case of high environmental consistency, all three algorithms can maintain stable tracking. The A-LOAM algorithm has the highest accuracy but poor stability. The pose estimation problem consists of multiple constraints, and the constraint strength in a certain direction will gradually decrease under the action of noise. In scenes with large scene changes, the tracking is unstable due to the poor quality of the collected feature points. The ORB-SLAM2 algorithm based on visual SLAM has the most obvious drift phenomenon. Compared with the ORB-SLAM2 algorithm, the LV-SLAM algorithm combines the depth information provided by the point cloud to obtain enough feature points with depth information and introduces the epipolar error to enhance the motion constraint, and the tracking is more stable. The detection range of lidar is long, and the effect is more obvious in the open environment. LV-SLAM has high adaptability to the environment, strong robustness, and high accuracy.

#### 3.1.2. Analysis of Positioning Accuracy

To further verify the improvement effect of the fusion algorithm compared with the visual SLAM algorithm, the absolute trajectory error (ATE), the relative pose error (RPE), and root-mean-square error (RMSE) of the algorithm output motion trajectory and the real trajectory are used as the main evaluation criterion for algorithm accuracy, among which ATE is also called APE in the Evo toolkit. In the evaluation process, the timestamps in the real motion trajectories are used as the benchmark, and the corresponding poses between different motion trajectories are determined by the timestamps. APE represents the sum of the Euclidean distances between the corresponding poses of different motion trajectories, which can directly reflect the overall accuracy of the algorithm. RPE refers to the Euclidean distance between adjacent poses corresponding to different trajectories, which mainly reflects the local accuracy of the algorithm. RMSE refers to the square root of the ratio of the square of the deviation between the estimated pose and the real pose and the ratio of the number of poses N, which measures the deviation between the predicted value and the true value and is more sensitive to outliers in the data, which can intuitively reflect the estimation and the degree of deviation of the motion trajectory.
(10)$$\left\{ \begin{array}{ccc} APE & = & \sqrt{{\left(x_{i}-{\bar{x}}_{i}\right)}^{2}+{\left(y_{i}-{\bar{y}}_{i}\right)}^{2}+{\left(z_{i}-{\bar{z}}_{i}\right)}^{2}}\\ RPE & = & \sqrt{{\left(x_{i}-x_{i-1}\right)}^{2}+{\left(y_{i}-y_{i-1}\right)}^{2}+{\left(z_{i}-z_{i-1}\right)}^{2}}-\sqrt{{\left({\bar{x}}_{i}-{\bar{x}}_{i-1}\right)}^{2}+{\left({\bar{y}}_{i}-{\bar{y}}_{i-1}\right)}^{2}+{\left({\bar{z}}_{i}-{\bar{z}}_{i-1}\right)}^{2}}\\ RMSE & = & \sqrt{\frac{1}{N}\sum_{i=1}^{N}{\left(x_{i}-{\bar{x}}_{i}\right)}^{2}+{\left(y_{i}-{\bar{y}}_{i}\right)}^{2}+{\left(z_{i}-{\bar{z}}_{i}\right)}^{2}} \end{array}\right.$$

The analysis of various indicators of the output error of the fusion SLAM algorithm and the visual SLAM algorithm in different scenarios is shown in Figure 8.

Figure 8 shows that in the feature-rich 00 sequence and 03 sequence, ORB-SLAM2 and LV-SLAM are almost the same. However, in the initial stage of ORB-SLAM2 algorithm initialization, the maximum error reaches 3.5 m. The A-loam algorithm is stable at the beginning of the operation, but the drift gradually increases with the travel distance. In the mixed suburban and urban scenarios, the data waves of all algorithms increase significantly, but the fusion algorithm and A-LOAM are more stable. In the 09 sequence, the feature points are relatively far away. Although the ORB-SLAM2 algorithm can detect the feature points, the optical flow does not change significantly, and the uncertainty of the depth information increases. Although the tracking can be maintained, the pose deviation is relatively small. When it is large, the trajectory drifts significantly. At the end of the scene, after the vehicle turns, the vision-based SLAM algorithm even loses tracking. However, LV-SLAM is not affected, and with the aid of the high-precision ranging of laser point clouds, the pose can still be accurately estimated.

The detailed information on the output trajectory error of the algorithm proposed in this paper and the visual SLAM algorithm and the lidar SLAM algorithm in the three scenarios are shown in Table 1.

Table 1 further quantifies the error size of the algorithm, including the maximum error (MAX), the minimum error (MIN), the root-mean-square error (RMSE), and the standard deviation (STD) of APE and RPE in each scenario. The bolded item is the algorithm error item with the lowest error in each index, and the performance of each error index of the two algorithms in each scene is different. Among them, the absolute trajectory and relative trajectory errors of ORB-SLAM2 are generally larger, especially in large-scale scenes such as 01 and 09, the operation effect is far less than that of the fusion SLAM algorithm, and even in scenes with fewer features, the loss of tracking results in mapping failure. The STD indicators of the fusion algorithm LV-SLAM are lower on the whole. The 03 sequence has the most abundant features, and the driving distance is 560.8 m. The errors of the LV-SLAM and ORB-SLAM2 algorithms are relatively small, and the RMSE is 0.5 m. Among them, the maximum trajectory error of LV-SLAM is 1.7 m. The MAX of ORB-SLAM2 is 3.6 m, which is 52.7% lower than the MAX of the pure visual SLAM algorithm. The total length of the 00 sequence is 3724.2 m, and the driving route is relatively complex. Multiple loops can be detected during the operation of the algorithm. There is no drift in general, but compared with the 03 sequence, the error increases and there are fluctuations. The RMSE of the LV-SLAM, ORB-SLAM2, and A-LOAM algorithms are increased by 258%, 409%, and 406%, respectively. The total length of the 09 sequence is 1705.05 m. Since the ORB-SLAM2 algorithm loses tracking when the vehicle returns to the starting point, the maximum trajectory error is as high as 116.07 m. The maximum error of LV-SLAM is only 5.4 m. The driving distance of the 09 sequence is three times that of the 03 sequence, but the vehicle finally returns to the starting point. Under the constraint of the loopback, the RMSE increases by 133.6% compared with the 03 sequence, which is only 2.29 m. The positioning accuracy of A-LOAM exceeds ORB-SLAM2 and LV-SLAM in both the 03 sequence and 09 sequence.

Overall, the LV-SLAM algorithm performs almost the same as the ORB-SLAM2 algorithm in feature-rich scenarios. However, as the number of effective features in the environment decreases, the errors of the two algorithms gradually increase, but the output pose error of the LV-SLAM algorithm is lower, and the operation is relatively stable. The short-range accuracy of A-loam is better than that of the LV-SLAM algorithm, but the LV-SLAM algorithm has better stability and accuracy at long distances. As the openness of the environment increases, the effect of BA is particularly pronounced in urban scenes with higher feature quality.

The LV-SLAM algorithm improves the stability of the algorithm by improving the objective function to enhance the motion constraints, combining the laser point cloud to obtain more effective feature points, and using the local optimization based on keyframes to improve the correlation between frames to achieve stable tracking and a high-precision positioning effect. Registration motion estimation and laser point cloud build high-precision dense maps to meet the requirements of navigation, obstacle avoidance, reconstruction, and other related work. In different types of scenarios, the effect of global non-drift and local optimality is achieved, and the algorithm has strong adaptability and high robustness.

### 3.2. Outdoor Scene Experiment

After the dataset is verified, to verify the effect of the fusion SLAM algorithm in the outdoor real environment, the experimental platform mentioned above is used to conduct real vehicle experiments. The mobile robot drives on the campus road. There are buildings and trees on both sides of the road, and the natural and structural features are mixed. At 4:00 p.m., an outdoor scene experiment was conducted on the university campus. The experimental site is shown in Figure 9.

Figure 9 shows the university campus from the aerial view of a satellite. The blue curve is the driving trajectory of the mobile robot, the yellow arrow is the movement direction of the robot, and the green dot is the starting point of movement, where the algorithm completes the initialization work. The red dot is the endpoint of movement. In the experiment, one person holds a laptop computer remotely connecting to the upper computer to run the algorithm, and the other person controls the chassis of the mobile robot to make a circuit. The mobile robot walks along the road marking line as much as possible. The motion is relatively gentle, which is convenient for analysis. To take advantage of the loopback, the mobile robot travels a certain distance past the starting point. In this experiment, the fusion algorithm LV-SLAM and the visual SLAM algorithm ORB-SLAM2 were used for testing. The experimental results are shown in Figure 10.

Figure 10 shows the dense map model output by the ORB-SLAM2 algorithm and LV-SLAM algorithm. Compared with the sparse feature point map of the visual SLAM algorithm, it contains more environmental information and can complete tasks such as navigation, obstacle avoidance, and 3D reconstruction. The red line is the motion trajectory of the mobile robot output by LV-SLAM. The trajectory is smooth and highly consistent with the map.

The motion trajectory obtained by the LV-SLAM algorithm and the ORB-SLAM2 algorithm is exported, and the motion trajectory is drawn using Origin. Since the Z-axis is perpendicular to the ground, only the two-dimensional trajectory map in the X and Y directions is drawn. The trajectory of the mobile robot is shown in Figure 11.

The black line and the red line in Figure 11 are the trajectories of the mobile robot output by the LV-SLAM algorithm and the visual SLAM algorithm ORB-SLAM2, respectively. Due to the limitation of the experimental conditions, the true value of the motion of the mobile robot cannot be obtained through the high-precision positioning system. However, it can be seen from the consistency of the motion trajectory of the mobile robot and the environment map shown in Figure 10b that the positioning effect of the LV-SLAM algorithm is more in line with the real situation. The ORB-SLAM2 algorithm has a sharp decrease in the number of keyframes in the area with reduced features, an increase in the pose estimation deviation, and a drift of the motion trajectory, which is offset by 30 m along the x-axis direction.

## 4. Conclusions

This paper proposes a new mobile robot positioning and mapping algorithm, LV-SLAM, which uses the data of the laser point cloud to provide additional depth information for image features to assist the monocular camera to complete motion estimation. The epipolar error is introduced, the residuals are calculated according to whether the feature points have depth inffformation, the objective function is reconstructed by the residuals of the feature points, and the pose is iteratively solved. The pose is optimized locally by the method based on keyframes, which reduces the complexity of the optimization problem and improves the accuracy of the algorithm. The LV-SLAM algorithm, the laser SLAM algorithm A-LOAM, and the visual SLAM algorithm ORB-SLAM2 were tested on the KITTI dataset and in outdoor real scenes. LV-SLAM has achieved good results in both mapping and positioning performance, and its performance is better than ORB SLAM2 and A-LOAM, these being single-sensing SLAM algorithms. The fusion algorithm LV-SLAM has good adaptability in different environments and has strong stability and high robustness.

In the future, we will consider fusing more sensors to improve the measurement accuracy of the algorithm. Multi-sensor fusion can make full use of the collected information to understand the position of the detected target. For example, combining with IMU to provide motion compensation can eliminate accumulated errors to a certain extent. We will further combine the semantic map to perceive the scene from both geometric and semantic aspects, so that the robot can have an abstract understanding of the environmental content and obtain advanced comprehensive information, thereby reducing the errors caused by dynamic scenes and improving the efficiency of loop closure detection.

## Figures and Tables

**Figure 1 sensors-22-04114-f001:**
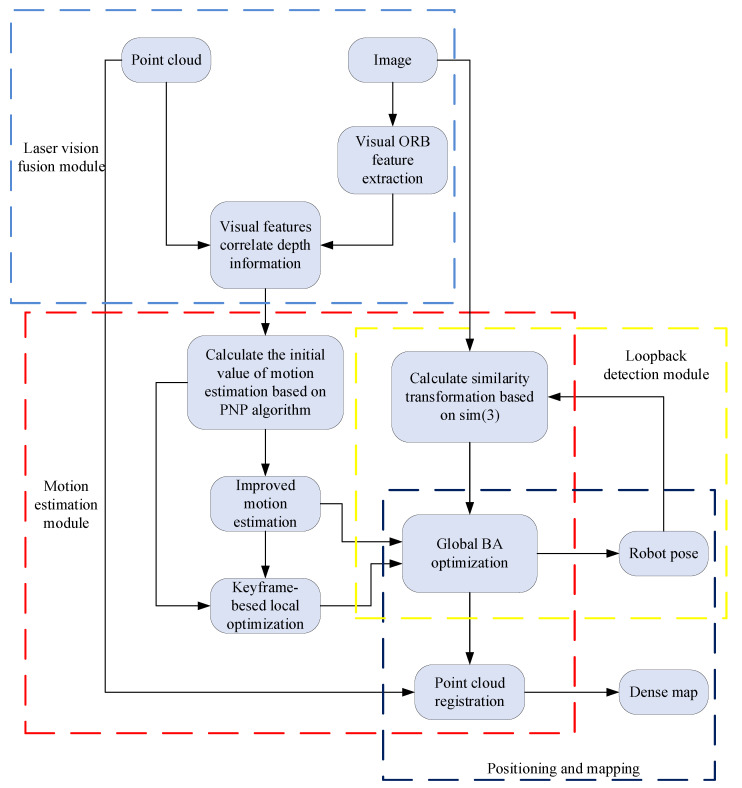
SLAM framework for image and laser point cloud fusion.

**Figure 2 sensors-22-04114-f002:**
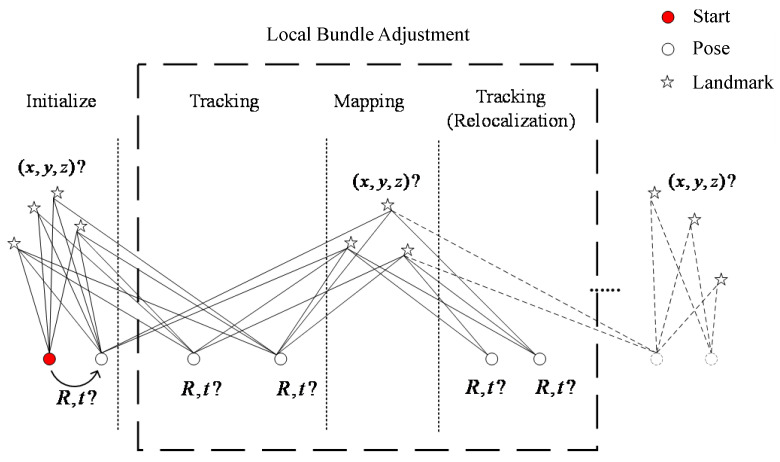
Feature-based V_SLAM local optimization.

**Figure 3 sensors-22-04114-f003:**
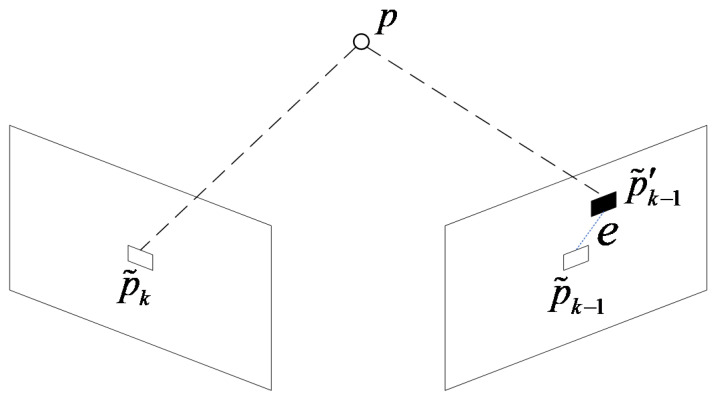
Schematic diagram of reprojection error.

**Figure 4 sensors-22-04114-f004:**
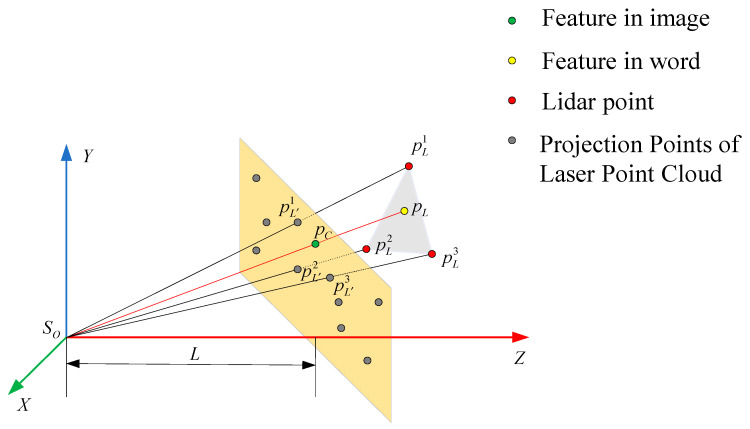
Schematic diagram of image feature correlation depth. The yellow plane represents the image normalization plane. The green dot represents the feature point of the image to be processed. The gray dot represents the laser point projected to the normalization plane. The red dot represents the laser point in the world coordinate system. The yellow dot represents the visual feature in the world coordinate system.

**Figure 5 sensors-22-04114-f005:**
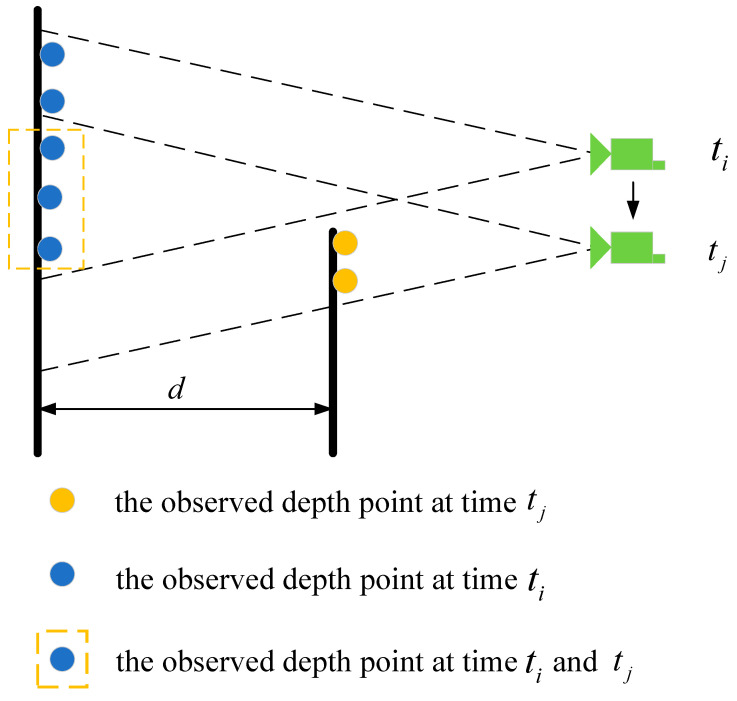
Feature depth validation.

**Figure 6 sensors-22-04114-f006:**
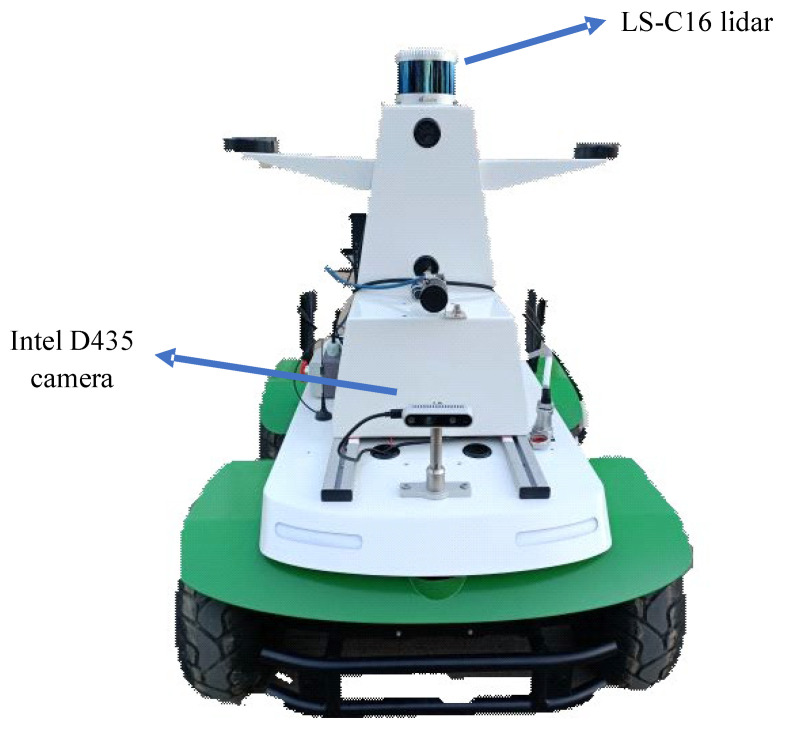
Experimental platform.

**Figure 7 sensors-22-04114-f007:**
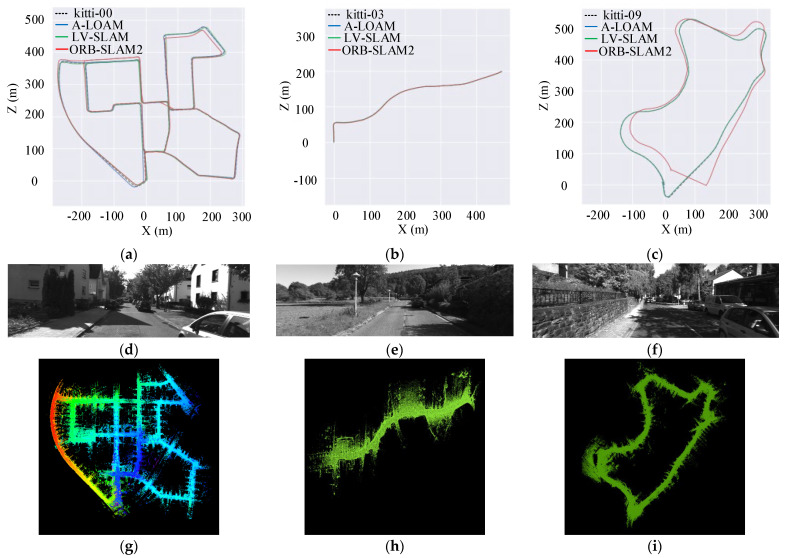
Comparison graph of each algorithm’s output trajectory. The datasets are chosen from three types of environments: urban, country, and suburb from left to right. In (**a**–**c**), we compare the results of the different algorithms to the GPS/INS ground truth. An image is shown from each dataset to illustrate the three environments in (**d**–**f**) and the corresponding point cloud map output by the LV-SLAM algorithm in (**g**–**i**).

**Figure 8 sensors-22-04114-f008:**
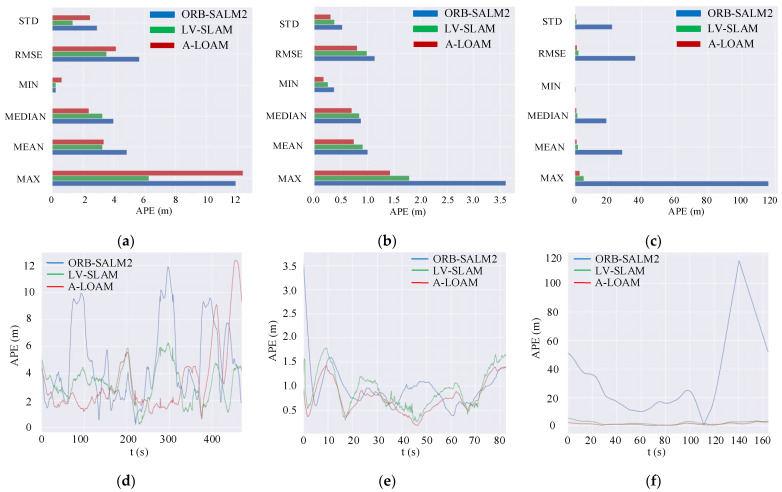
Comparison of trajectory errors of KITTI sequences. The datasets are chosen from three types of environments: urban, country, and suburban from left to right. In (**a**–**c**), we compare the kinds of error indicators of the fusion SLAM algorithm LV-SLAM to the visual SLAM algorithm and the lidar SLAM algorithm, including the maximum error (MAX), the mean error (MEAN), the median error (MEDIAN), the minimum error (MIN), the root-mean-square error (RMSE), and the standard deviation (STD). The corresponding absolute error of the motion trajectory is a function of travel time in (**d**–**f**).

**Figure 9 sensors-22-04114-f009:**
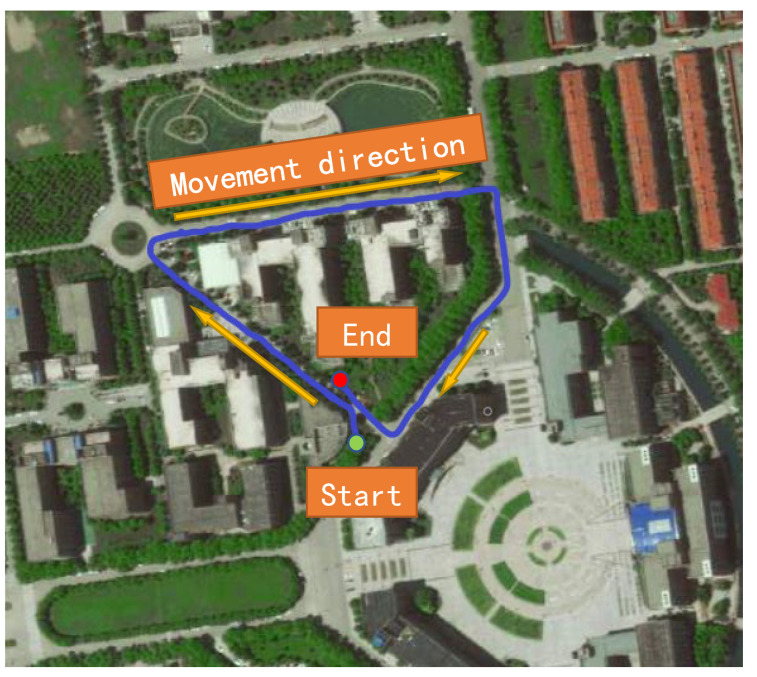
Outdoor real experimental scene.

**Figure 10 sensors-22-04114-f010:**
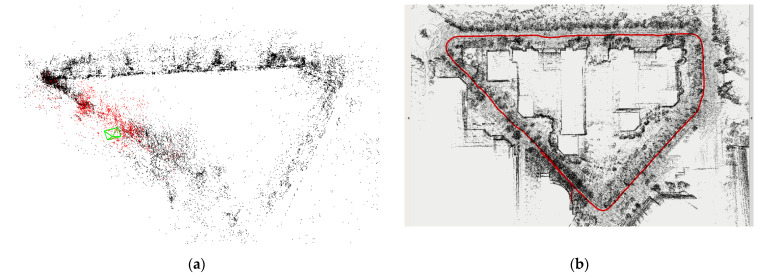
Mapping effect of ORB-SLAM2 algorithm and LV-SLAM algorithm. (**a**) The map output by the ORB-SLAM2 algorithm; (**b**) the map output by the LV-SLAM algorithm.

**Figure 11 sensors-22-04114-f011:**
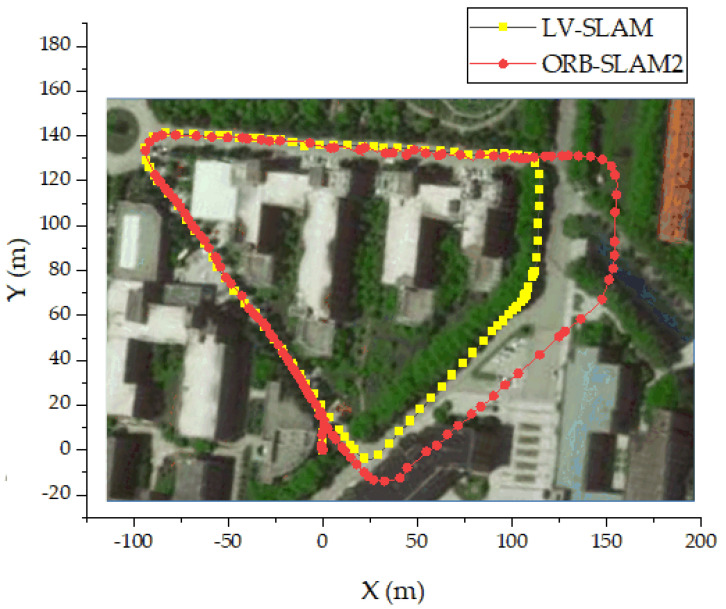
The trajectory of the mobile robot.

**Table 1 sensors-22-04114-t001:** KITTI sequence error analysis.

Scenes		Algorithm	MAX (m)	MIN (m)	*RMSE* (m)	STD (m)
00 Sequence	APE	ORB-SLAM2	11.8961	0.225525	5.65063	2.90775
		LV-SLAM	6.2644	0.231221	3.51915	1.32272
		A-LOAM	12.3746	0.606901	4.14547	2.45925
	RPE	ORB-SLAM2	4.89548	0.00359534	0.212906	0.180132
		LV-SLAM	0.872029	0.00399336	0.0739505	0.0443978
		A-LOAM	9.62911	0.00850367	2.87122	2.12342
03 Sequence	APE	ORB-SLAM2	3.60416	0.377288	1.13878	0.529162
		LV-SLAM	1.79031	0.260234	0.989652	0.382163
		A-LOAM	1.42551	0.178019	0.806478	0.30732
	RPE	ORB-SLAM2	0.186905	0.00551481	0.0588605	0.035607
		LV-SLAM	0.828702	0.00223307	0.0624783	0.0462032
		A-LOAM	3.81938	0.146385	1.82442	0.684952
09 Sequence	APE	ORB-SLAM2	116.065	0.625562	36.3116	22.5404
		LV-SLAM	5.40621	0.335209	2.29366	1.12238
		A-LOAM	3.08308	0.107195	1.37112	0.617775
	RPE	ORB-SLAM2	174.979	0.201846	6.81681	6.79056
		LV-SLAM	0.32931	0.0572069	0.0712657	0.0315476
		A-LOAM	6.41395	0.390049	2.61259	1.04912

## Data Availability

The datasets used or analyzed during the current study are available from the corresponding author on reasonable request.

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
