# Peer review of "Mobile Robot Localization and Mapping Algorithm Based on the Fusion of Image and Laser Point Cloud"

_sensors, 2022, doi:10.3390/s22114114_

Round 1
Reviewer 1 Report
- Define the non-standard abbreviations before using the article. For example LC-fusion, i believe it stands for laser-camera-fusion.
- Literature review is well presented but define the novelty/originality of the proposed method in comparison to the literature.
- The article needs to be understandable for a wide range of audience. So define and cite the terms appropriately. For example, KITTI dataset is first used in the introduction but it is defined in 3.1. A detailed description is not required but it would be better to introduce the terms with simple statements.
- A detailed theoretical model is presented in Section 2. My suggestion would be remove the content that is already available in the literature and focus on the originality of your proposed method. define the original concept used in the article. As the terms available in the literature can briefly explained with proper citation.
- The concept stability over time is mentioned many times. Explain the reason for the decrease in stability over time for example with A-LOAM as mentioned in Section 3.1.1
- In the conclusion, it is mentioned that LC-Fusion, ORB-SLAM2 and A-LOAM are tested/compared but mainly ORB-SLAM2 and LC-Fusion are compared except in section 3.1.1.
- Is it not general that laser and camera based SLAM will always have better results than vision based ORB-SLAM2. Please explain the motivation in comparing your method with vision based approach.
- Clearly define that originality/novelty of the article is a demonstration or the method/model/image processing.
Author Response
- Define the non-standard abbreviations before using the article. For example LC-fusion, i believe it stands for laser-camera-fusion.
Answer: Thanks for your detailed review. Considering this suggestion, we have revised all the non-standard abbreviations LC-fusion to be LV-SLAM, shown in the article.
- Literature review is well presented but define the novelty/originality of the proposed method in comparison to the literature.
Answer: Thank you very much for your valuable recommendation. We briefly summarize the characteristics and novelties of our method in the introduction.
- The article needs to be understandable for a wide range of audience. So define and cite the terms appropriately. For example, KITTI dataset is first used in the introduction but it is defined in 3.1. A detailed description is not required but it would be better to introduce the terms with simple statements.
Answer: Thanks for your valuable review. We have cited the terms KITTI dataset with a simple statement in the Introduction section.
- A detailed theoretical model is presented in Section 2. My suggestion would be remove the content that is already available in the literature and focus on the originality of your proposed method. define the original concept used in the article. As the terms available in the literature can briefly explained with proper citation.
Answer: Thanks for your valuable suggestion, you are right. We have simplified the model in Section 2, which focuses on expressing innovative modules.
- The concept stability over time is mentioned many times. Explain the reason for the decrease in stability over time for example with A-LOAM as mentioned in Section 3.1.1
Answer: Thanks for your detailed review. We have added an explanation after this conclusion.
- In the conclusion, it is mentioned that LC-Fusion, ORB-SLAM2 and A-LOAM are tested/compared but mainly ORB-SLAM2 and LC-Fusion are compared except in section 3.1.1.
Answer: Thanks for your detailed review. we have added a direct, quantitative comparison between LC-Fusion and A-LOAM in Section 3.1.2.
Moreover, we have reanalyzed the experimental results and revised the statement in Section 3.1.2.
For outdoor real experiments, unlike the KITTI dataset's radar which is a 64-line lidar, our experiments use a 16-line lidar. In the case of sparse point clouds, the A-LOAM algorithm we run in the selected scene often loses tracking and thus fails to build the map. So we did not consider the A-LOAM algorithm when we compared the algorithm performance.
- Is it not general that laser and camera based SLAM will always have better results than vision based ORB-SLAM2. Please explain the motivation in comparing your method with vision based approach.
Answer: Thanks for your detailed review. Under the same experimental conditions and the same camera sensor, the LV-SLAM algorithm can achieve better global results than ORB-SLAM2. The denser the laser point cloud, the better the localization effect will be.
The algorithm in this paper is optimized and improved in a vision-based framework. The laser point cloud is fused to complete the positioning and mapping work based on the image features.
- Clearly define that originality/novelty of the article is a demonstration or the method/model/image processing.
Answer: Thanks for your valuable review. We have stated that the originality/novelty of the article is a localization and mapping method in the introduction.
Special thanks for your valuable suggestions.
Reviewer 2 Report
(1)The content of the current version is redundancy. Though it is for its understandable to the readers, some description should be condensed. Please refer to the standard for article of science and technology.
(2)The main contributions (or innovation) and the motivations of this study are still confused.
(3)In the revised manuscript, the theorem and proof are added. However, the writing does not meet the standard and mathematical form. Please improve it in the further revision.
Author Response
- The content of the current version is redundancy. Though it is for its understandable to the readers, some description should be condensed. Please refer to the standard for article of science and technology.
Answer: Thank you very much for your valuable suggestion. We have condensed some redundant descriptions.
And we have added the citations for the terms available in the literature with a simple statement.
- The main contributions (or innovation) and the motivations of this study are still confused.
Answer: Thanks for your valuable review. We briefly summarize the characteristics and novelties of our method in the introduction.
Answer: We are very sorry for our incorrect writing. Considering this suggestion, we have modified the Equation (5) in Section 2.2.
And modified Equation (8) in Section 2.4.
Special thanks for your valuable suggestions.
Please see the attachment
Reviewer 3 Report
In this study, the authors introduced a motion estimation algorithm that enhances the monocular camera with laser-based depth information. Using a pure vision-based algorithm and a pure laser-based algorithm as the benchmark, the proposed algorithm demonstrated lower localization error when tested on selected subset of the KITTI datasets and a preliminary real-world testing (compared with the vision-based algorithm). Overall, the manuscript is well-written with some minor typos and grammar issues. Some control/benchmarks are missing (details below). The quality of the manuscript can be improved by addressing the items below:
- The most important issue is the choice of benchmark. In Fig. 7, authors compared LC-Fusion with A-LOAM (laser only) and ORB (vision only) qualitatively. And it seems that LC-Fusion and A-LOAM performed equally well (line 377). However, on line 379-380, the authors claimed that A-LOAM has poor stability and can lose track on long straight sections, which is not supported by results shown in Fig. 7 at all.
- Following up on point 1, in subsequent quantification of errors for algorithm comparisons (Fig. 8 and table 1), the authors only compare LC-Fusion with ORB and completely excluded A-LOAM. While the LC-Fusion indeed outperformed ORB, how about a direct, quantitative comparison between LC-Fusion and A-LOAM?
- Another follow-up on points 1 and 2:the authors only compare LC-Fusion with pure-vision and pure-laser algorithms, other similar vision-laser SLAM algorithms were not considered and compared. Specifically, can you provide a direct comparison between LC-Fusion and another existing vision-laser SLAM algorithm?
- How is the proposed LC-Fusion method different from existing vision-laser SLAMs? You can make this clear in the introduction section.
- Figure caption typos and errors: line 225-226: "the yellow plane represents xxx " was repeated. Some elements were not included in the explanation, e.g., green dot and yellow dot.
- Error: line 232: "the blue plane", should this be 'the gray plane'?
- One line 261: how is 'd' determined?
- Typo: line 294: "Online 13" -->"On line 13"
- Line 481-482: "Due to the limitation of experimental conditions, the true value of the motion of the 481
mobile robot cannot be obtained through the high-precision positioning system. " Does this mean that the true motion/GPS/true coordinates are missing? What is the reason for this "limitation of experimental conditions" that prevent the usage of GPS data? - It is hard to see that LC-Fusion is better than ORB on Fig. 10. please add a overlay of the campus map to both a and b.
- Fig. 11: Same for this, adding an overlay of the campus map to this figure to actually show that LC-Fusion is closer to the ground truth.
Author Response
- The most important issue is the choice of benchmark. In Fig. 7, authors compared LC-Fusion with A-LOAM (laser only) and ORB (vision only) qualitatively. And it seems that LC-Fusion and A-LOAM performed equally well (line 377). However, on line 379-380, the authors claimed that A-LOAM has poor stability and can lose track on long straight sections, which is not supported by results shown in Fig. 7 at all.
Answer: We are sorry for our negligence. This conclusion that A-LOAM has poor stability and can lose track on long straight sections was found by my usual experiments, and now we have deleted it.
- Following up on point 1, in subsequent quantification of errors for algorithm comparisons (Fig. 8 and table 1), the authors only compare LC-Fusion with ORB and completely excluded A-LOAM. While the LC-Fusion indeed outperformed ORB, how about a direct, quantitative comparison between LC-Fusion and A-LOAM?
Answer: Considering this suggestion, We have added a direct, quantitative comparison between LC-Fusion and A-LOAM in Section 3.1.2.
Moreover, we have reanalyzed the experimental results and revised the statement in Section 3.1.2.
For outdoor real experiments, unlike the KITTI dataset's radar which is a 64-line lidar, our experiments use a 16-line lidar. In the case of sparse point clouds, the A-LOAM algorithm we run in the selected scene often loses tracking and thus fails to build the map. So we did not consider the A-LOAM algorithm when we compared the algorithm performance.
- Another follow-up on points 1 and 2:the authors only compare LC-Fusion with pure-vision and pure-laser algorithms, other similar vision-laser SLAM algorithms were not considered and compared. Specifically, can you provide a direct comparison between LC-Fusion and another existing vision-laser SLAM algorithm?
Answer: Thanks for your valuable suggestion. The original intention of this article is to optimize and improve the visual SLAM algorithm, and the structure of the article is arranged accordingly. Please think about it. If you still insist that the article should provide a direct comparison between LC-Fusion and another existing vision-laser SLAM algorithm, I'm still willing to deal with it in the next version.
- How is the proposed LC-Fusion method different from existing vision-laser SLAMs? You can make this clear in the introduction section.
Answer: Thank you very much for your valuable recommendation. We briefly summarize the characteristics and novelties of our method, comparing it with the existing vision-laser SLAM algorithm, LIMO, in the introduction.
- Figure caption typos and errors: line 225-226: "the yellow plane represents xxx " was repeated. Some elements were not included in the explanation, e.g., green dot and yellow dot.
Answer: Thanks for your detailed review. We are sorry for our negligence. We have removed duplicates and explained the missing part of Figure 4.
- Error: line 232: "the blue plane", should this be 'the gray plane'?
Answer: We are sorry for our negligence. We have revised the statement.
- One line 261: how is 'd' determined?
Answer: Thanks for your detailed review. 'd' is Customized based on test results. And it is set to be 2m in the algorithm program.
- Typo: line 294: "Online 13" -->"On line 13"
Answer: We are very sorry for our incorrect writing. we have revised "Online 13" to be "On line 13".
- Line 481-482: "Due to the limitation of experimental conditions, the true value of the motion of the mobile robot cannot be obtained through the high-precision positioning system. " Does this mean that the true motion/GPS/true coordinates are missing? What is the reason for this "limitation of experimental conditions" that prevent the usage of GPS data?
Answer: We are very sorry for our unclear expression. I mean there are some problems with the hardware of the test equipment, and we can't get enough accurate and stable GPS data with it. However, the reliability of the positioning accuracy can still be reflected by the consistency of the trajectory and the map. We have revised the statement.
- It is hard to see that LC-Fusion is better than ORB on Fig. 10. please add a overlay of the campus map to both a and b.
Answer: Thanks for your valuable suggestion, we have to explain to you Figure 10 is intended to show that the LV-SLAM algorithm can output a high-precision dense map. And Fig. 10(a)(b) are screenshots of the algorithm running results, which cannot add an overlay of the campus map to both a and b.
- Fig. 11: Same for this, adding an overlay of the campus map to this figure to actually show that LC-Fusion is closer to the ground truth.
Answer: Thank you very much for your valuable suggestion Considering this suggestion. We have added an overlay of the campus map to Figure 11.
Special thanks for your valuable suggestions.
Please see the attachment
Round 2
Reviewer 2 Report
The commnets are tackled and the manuscript can be accepted after the minor revisions, such as the grammatical mistakes.
Reviewer 3 Report
Thank you for taking to time to address the review report and I think the quality of the manuscript has been significantly improved.